# Tramesan Elicits Durum Wheat Defense against the Septoria Disease Complex

**DOI:** 10.3390/biom10040608

**Published:** 2020-04-14

**Authors:** Valeria Scala, Chiara Pietricola, Valentina Farina, Marzia Beccaccioli, Slaven Zjalic, Fabrizio Quaranta, Mauro Fornara, Marco Zaccaria, Babak Momeni, Massimo Reverberi, Angela Iori

**Affiliations:** 1Consiglio per la Ricerca in Agricoltura e l’Analisi dell’Economia Agraria, Centro di Ricerca Difesa e Certificazione, Via C.G. Bertero, 22, 00156 Roma, Italy; valeria.scala@crea.gov.it; 2Università Sapienza, Dip. Biologia Ambientale, P.le Aldo Moro 5, 00185 Roma, Italy; chiara.pietricola@live.it (C.P.); valentina.farifari@gmail.com (V.F.); marzia.beccaccioli@uniroma1.it (M.B.); 3Department of Ecology, Agronomy and Aquaculture, University of Zadar, Ulica Mihovila Pavlinovića bb, 23000 ZADAR, Croatia; szjalic@unizd.hr; 4Consiglio per la Ricerca in Agricoltura e l’Analisi dell’Economia Agraria, Centro di ricerca Ingegneria e Trasformazioni agroalimentari, Via Manziana 30, 00189 Roma, Italy; fabrizio.quaranta@crea.gov.it (F.Q.); mauro.fornara@crea.gov.it (M.F.); angela.iori@crea.gov.it (A.I.); 5Department of Biology, Boston College, Chestnut Hill, MA 02467, USA; zaccarim@bc.edu (M.Z.); momeni@bc.edu (B.M.)

**Keywords:** biostimulant, plant defense, mushrooms, antioxidant, Septoria disease complex, wheat

## Abstract

The Septoria Leaf Blotch Complex (SLBC), caused by the two ascomycetes *Zymoseptoria tritici* and *Parastagonospora nodorum*, can reduce wheat global yearly yield by up to 50%. In the last decade, SLBC incidence has increased in Italy; notably, durum wheat has proven to be more susceptible than common wheat. Field fungicide treatment can efficiently control these pathogens, but it leads to the emergence of resistant strains and adversely affects human and animal health and the environment. Our previous studies indicated that active compounds produced by *Trametes versicolor* can restrict the growth of mycotoxigenic fungi and the biosynthesis of their secondary metabolites (e.g., mycotoxins). Specifically, we identified Tramesan: a 23 kDa α-heteropolysaccharide secreted by *T. versicolor* that acts as a pro-antioxidant molecule in animal cells, fungi, and plants. Foliar-spray of Tramesan (3.3 μM) on SLBC-susceptible durum wheat cultivars, before inoculation of causal agents of Stagonospora Nodorum Blotch (SNB) and Septoria Tritici Blotch (STB), significantly decreased disease incidence both in controlled conditions (SNB: −99%, STB: −75%) and field assays (SNB: −25%, STB: −30%). We conducted these tests were conducted under controlled conditions as well as in field. We showed that Tramesan increased the levels of jasmonic acid (JA), a plant defense-related hormone. Tramesan also increased the early expression (24 hours after inoculation—hai) of plant defense genes such as PR4 for SNB infected plants, and RBOH, PR1, and PR9 for STB infected plants. These results suggest that Tramesan protects wheat by eliciting plant defenses, since it has no direct fungicidal activity. In field experiments, the yield of durum wheat plants treated with Tramesan was similar to that of healthy untreated plots. These results encourage the use of Tramesan to protect durum wheat against SLBC.

## 1. Introduction

Wheat is the main source of plant proteins in human diet. Plant diseases such as rusts, head blight, powdery mildew, and leaf blotch negatively affect wheat yield. Leaf blotch is caused by fungal pathogens, including *Parastagonospora nodorum* and *Zymoseptoria tritici* [1,2], and can significantly damage the global food security [3]. Septoria Leaf Blotch Complex (SLBC), one of the main diseases of wheat, may cause up to 50% crop loss in years of severe epidemics [4].

*Parastagonospora nodorum* is a necrotrophic pathogen of both common and durum wheat. It causes necrotic leaf spots and thus hampers photosynthesis. To infect the plant, *P. nodorum* produces several necrotrophic effectors; Tox1, Tox3, and ToxA are small secreted proteins that alter or suppress the host immune response [5]. These effectors interact in a reverse manner with respect to the “gene-for-gene” model, with corresponding sensitivity loci in wheat [6,7]. *Zymoseptoria tritici* is a hemibiotroph—parasitic in living tissue and persistent in dead tissue—that commonly undergoes sexual reproduction; local populations are extremely variable and can rapidly adapt to fungicides [8]. By causing leaf necrosis, *Z. tritici* reduces the grain-filling potential of wheat and, as a consequence, yield.

Recently, the molecular basis of the interaction of wheat (specifically common wheat) with SLBC has been under the spotlight [9,10]. *P. nodorum* and *Z. tritici* infections account for approximately 70% of annual cereal fungicide (>€400 million) usage in the European Union (EU) [3]. To counter these pathogens, chemical fungicides such as strobilurins, triazoles, and imidazoles are widely used; however, resistant fungal strains have been emerging [11]. The European regulation 128/2009 governing the use of pesticides [12] has recently removed several fungicides from the market to contain the environmental and health-related issues. At present, the employment of non-synthetic (natural) compounds in agriculture is increasing worldwide [13]: e.g., volatile molecules ((2,6-dichloroisonicotinic acid (INA), benzothiadiazole (BTH) commercialized as BION^®^); mineral nutrient and non-protein amino acids such as DL-3-amino-n-butanoic acid (BABA); cellular or molecular (Pathogen/Microbe/Damage Associated Molecular Patterns) PAMP, MAMP, or DAMP [14]. The mechanism behind resistance is presumably based on the induction of the host innate immune response. These responses are associated with the direct induction of antimicrobial proteins [15] via systemic acquired resistance, induced systemic resistance (ISR), and mycorrhiza-induced resistance. At the same time, a direct induction is not the only way to enhance plant defense. Defense priming is a viable approach to “boosted responses against pathogens” induced at a systemic level [16]. Since adaptive plasticity prevents an immune response in the absence of a challenge [16], defense priming allows the immune system to respond to stimuli with little energy expenditure. Defense priming responses are highly effective [17] and challenge-specific [18]. Plants can be “primed” with select elicitors to counter future attacks more efficiently than non-primed plants [19].

In this study, we explore the ability of purified Tramesan, a 23 kDa α-heteropolysaccaride secreted by the basidiomycete *Trametes versicolor* [20], to act as an elicitor of durum wheat defenses for SLBC. Tramesan has demonstrated biological activity on fungal, plant, and animal cells [19,20]. Tramesan appears to act as a pro-antioxidant by inducing the expression of oxidative stress response-related genes, thus triggering a biological response. In *Aspergillus flavus*, a mycotoxigenic fungus, the presence of Tramesan strongly inhibits aflatoxin biosynthesis, without significantly influencing fungal growth. Aflatoxin biosynthesis is a response to oxidative stress in fungal cells [21]. The presence of Tramesan enhances the expression of *AfyapA*, an oxidative stress-related transcriptional factor, thus maintaining oxidant/antioxidant balance and downregulating aflatoxin synthesis. Similarly, Tramesan enhances the antioxidant response in wheat leaf cells providing better resistance to some fungal pathogens [19]. In murine melanoma cell lines, in which intrinsic oxidative stress boosts cell division, the presence of Tramesan resulted in reduction of cell growth and enhanced production of melanin [22].

In light of the data presented this work, we propose that Tramesan significantly reduces SBLC severity by eliciting durum wheat innate defense.

## 2. Materials and Methods

### 2.1. Preparation of Fungal Inoculum 

The fungal pathogen *Parastagonospora nodorum*, strain 15465, was isolated at the Council for Agricultural Research and Economics—Research Centre for Engineering and Agro-Food processing, CREA-IT, and submitted to the Culture Collection Agri-Food Important Toxigenic Fungi-Item, Institute of Science of Food Production (ISPA), National Research Council (CNR), Bari, Italy [23]. The fungus, stored at −20 °C, was revitalized by transferring it into fresh potato dextrose agar plates (41g/L PDA, HiMedia, Mumbai, India) and incubated at 20 °C with a photoperiod of 12 h over 10 days. After growth, conidia were transferred with sterile needles to PDA Petri dishes and placed in a growth cabinet under the same temperature and light conditions. After incubation, sterile distilled water was added to each plate and the spores were gently scraped with special glass rods. The obtained spore suspension was filtered with a sterile gauze and diluted to a concentration of 10^6^ spores/mL [24]. Tween 20 surfactant was added to the suspension (0.1% v/v) to increase the adhesion to leaves.

*Zymoseptoria tritici*, strain 18258 was isolated from durum wheat at CREA-IT. To test its virulence, wheat cultivars (durum wheat: Svevo and Duilio cvs; bread wheat: Salamandra cv) were inoculated with Zt 18258 in a greenhouse environment [25]. After 4 days, leaf segments from seedlings were cut and put on agar in Petri dishes at 20 °C for subsequent culture and monoconidial isolation, identification, and storage. Three weeks later, a high presence of pycnidia was observed. The Zt strain was pathogenic to both wheat species. It was revitalized by transferring it to fresh PDA medium (potato dextrose agar, 39 g/L HiMedia, Mumbai, India) and incubated at 18 °C with a photoperiod of 12 h over 10 days. After incubation, sterile distilled water was added to each plate and spores were scraped gently with special glass rods. The spores were then transferred to yeast–sucrose liquid medium (yeast extract 10 g/L; sucrose 10 g/L) and left shaking for 7 days at 18 °C, with permanent light. The spores were collected by centrifugation at 5000 rpm for 5 min at 15 °C, washed twice with sterile distilled water, and resuspended in a MgSO_4_ solution (0,01 M), containing Tween 20 surfactant (0.1% v/v). The concentration was adjusted to 10^7^ spores/mL.

### 2.2. Tramesan Preparation 

*Trametes versicolor*, strain C, [26] is regularly maintained on potato dextrose agar (PDA, Difco) at 4 °C, and renewed every 60 days at the collection of the Rome Botanical Garden. *T. versicolor* was grown for 7 days on PDA in Petri dishes at 25 °C. After 7 days, 3 plugs of 1 cm diameter, with uniformly grown biomass (mycelium + medium), were collected and added, under sterile conditions, to 100 mL of potato dextrose broth (PDB) and agitated at 150 rpm, at 25 °C, over 14 days. The entire mass (liquid medium + mycelium) was then homogenized using a Waring blender (speed 5 for 3 pulses of 2 min each) and added at 5% v/v to PDB medium and grown over 14 days. Tramesan was purified as described by [20], then 2.3 g of lyophilized Tramesan were dissolved in 1 L of distilled sterile water. This solution was used to spray wheat plants in a greenhouse environment. Plants were treated with Tramesan at 3.3 μM as previously described [20]. Tramesan was formulated with 0.1% v/v of Tween 20 as surfactant to increase its adhesion to leaves.

### 2.3. Trials in Phytotron

Kernels of two Italian commercial varieties of durum wheat (Svevo and Duilio; Syngenta Italia and SIS Società Italiana Sementi, respectively, *T. turgidum* subsp. *durum* (Desf. Husn.) moderately susceptible to SLBC, were superficially disinfected with a sodium hypochlorite solution (0.1% v/v, 10 minutes agitation) and washed three times with sterile distilled water. The disinfected seeds were placed in Petri dishes containing water/agarose (2% w/v) and incubated for 24 h at 20 °C, followed by 24 h at 4 °C, and finally 48 h at 20 °C for germination. The kernels were transferred in a twice autoclaved (20 min at 121 °C) soil mixture (20 L of soil/5 L of perlite), in pots of 0.5 L. In a phytotron, temperature, humidity, and light were regulated to 20 °C, 80% humidity, and 18 h light exposure with 150 μmol of photon m^−2^s^−1^ from 6 am to 10 pm. The pots were positioned on a rotary floor, so that different plants were subject to the same conditions. Plants (*n* = 48; 24 var. Svevo, 24 var. Duilio for each experimental condition) were irrigated three times a week. Four different experimental conditions were used: (1) plants non-inoculated and non-treated with Tramesan (Mock); (2) plants treated with Tramesan (T); (3) plants inoculated with *P. nodorum* ST15465 or *Z. tritici* ST18258 (Inf); (4) plants treated and inoculated with *P. nodorum* ST15465 or *Z. tritici* ST18258 (T + Inf). Tramesan (100 mL per 64 pots; 3.3 μM) was sprayed on seedlings of the 2 varieties at the 2–3 leaf stage (second leaf fully expanded), 48 h prior to pathogen inoculation. The inoculation of wheat plants with foliar pathogens was carried out by spraying conidial suspensions (*P. nodorum*: 10^6^ spores/mL; *Z. tritici*: 10^7^ spores/mL) containing Tween 20 surfactant (0.1% v/v) as previously described [24]. The inoculated plants were covered with black plastic sheets for 24 h and with transparent plastic sheets in the subsequent 24 h, providing a humid saturated atmosphere necessary for spore germination and penetration into the leaves. Trials were conducted on a 9-day observation period. To assess Stagonospora Nodorum Blotch (SNB) or Septoria Tritici Blotch (STB) symptoms, a severity index (numerical rating from 1 to 5) was adopted following Liu’s scale [27], where 0 represents absence of symptoms and 5 indicates large coalescent lesions. Results from this scale were then transformed into percentage of lesioned area (as reported by Liu) using the arc sen √ (x) formula. To study the early effect of Tramesan on infected wheat, the seedling leaves were collected for subsequent analysis from −48 h to 9 days after infection (dai).

### 2.4. In-field Artificial Infection Trials 

Field trials in the growing season 2014–2015 were performed in Montelibretti (CREA-IT) and in Rome using durum wheat variety Svevo and the same strains of *P. nodorum* ST15465 and *Z. tritici* ST18258 used for artificial inoculations (see Section 2.3). In the crop season 2014–2015, we have 12 plots in two different fields located in Rome (latitude 41.969277; longitud 12.464256; m.a.s.l. 20) and in Montelibretti (RM) (latitude 42.129287; longitude 12.63969; m.a.s.l. 25) at the CREA–Research Centre for Engineering and Agro-Food Processing, CREA-IT. The Svevo wheat variety was sown (viable seeds: 450/m^2^) in plots of 1.5 m × 2 m (3 m^2^) using a randomized block design with three replicates. Different experimental conditions were set up (Table 1).

Each plot in the two fields contained *n* = 250 plants. The number of plants was fewer than the seeds, probably due to the considerable presence of birds at the time of sowing. The trials were performed in three plots per thesis (mock, T, Inf, T + Inf). Inoculation for the artificial contamination in the field plots was prepared as indicated in Section 2.1. As in Table 2, the plants were treated with Tramesan by spraying at the booting growth stage, GS 47, for SNB trials, and at stem elongation growth stage, GS 37, for STB trials. The inoculation of wheat plants with foliar pathogens was carried out by spraying conidial suspensions (*P. nodorum*: 10^6^ spores/mL at growth stage GS 49; *Z. tritici*: 10^7^ spores/mL at growth stage GS 39) containing Tween 20 surfactant (0.1% v/v) as previously described [22]. The inoculated plots were covered with transparent plastic sheets for 48 h, providing a humid, saturated atmosphere necessary for spore germination and penetration into the leaves. Over that plastic sheet, a plastic shade was used to protect the plants from direct sunlight. Infections were assessed by visual observations of symptoms, such as STB and SNB severity on the flag leaf at growth stage GS 83, according to Iori et al., 2015 [24]. In the field, diseases were assessed every 7 to 10 days until the ripening stages. All field trials were harvested at maturity.

### 2.5. Calibration Curve of P. nodorum and Z. tritici and Quantification of Fungal DNA by Real-time PCR

Total DNA was extracted from spores of *P. nodorum* ST15465 and of *Z. tritici* ST18258. The spores (10^8^) were collected by pipetting 50 μL of Triton-X100 (Merck KGaA, Darmstadt, Germany) in Petri dishes (9 cm) completely covered with conidiating mycelia. Then, 500 μL of cetyl trimethylammonium bromide (CTAB) buffer (1 M Tris at pH 8, 5 M NaCl, 0.5 M EDTA, 2% w/v CTAB) and glass beads (425–600 μm; Sigma–Aldrich) were added to the spores/Triton-X100 mix in 2.0 mL tubes and DNA was extracted as previously described [24]. Real time PCR was performed in a LineGene K PCR detection system (Bioer, Hangzhou, China) with the following cycling conditions: 95 °C for 10 min, followed by 40 cycles of 95 °C for 15 s, 55 °C for 30 s, and 72 °C for 1 min to amplify the β-tubulin gene (for_TGGGTACGCTTTTGATCTCC; rev_AACGAGGTGGTTCAGGTCAC) of *Parastagonospora nodorum* (acc. n. KF252679). Cycling of 95 °C for 10 min, followed by 40 cycles of 95 °C for 15 s, 62 °C for 30 s, and 72 °C for 1 min to amplify the GAPDH gene (for_TCCGTCGTTGACTTGACCTG; rev_TCTGAACTCAACGGTCGCTT) of *Zymoseptoria tritici* (acc. n. XM_003855615). Standard calibration was performed plotting the real-time PCR (RT-PCR) signals obtained for *S. nodorum* and *Z. tritici* genomic DNA extracted from fungal strains in the concentration range of 5 pg to 50 ng. The equation describing the increase of DNA concentration was calculated (y = 3 × 10^8^e − 0.6396x, R^2^ = 0.9967 for *P. nodorum*; y = 2 × 10^6^e − 0.756x, R^2^ = 0.9864 for *Z. tritici*) and used afterwards as a reference standard for the extrapolation of quantitative information for DNA targets of unknown concentrations. The efficiency of the PCR reaction (102%) was obtained from the calibration curves slope (E = 10^−1/slope^ −1). For quantifying the fungal amount within the plant tissues, total DNA was extracted from the wheat leaves (*n* = 2 leaves/plants; *n* = 196 plants) at different times after pathogen inoculation (0–9 days after inoculation; dai) according to the Färber method with minor modifications [28]. RT-PCR (Bioer, Hangzhou, China) as indicated above used total DNA extracted from wheat leaves for PCR quantification of the amount of fungal DNA within plant tissues. RT-PCR amplification reactions were carried out in three technical replicates, and the results were expressed as ng of target DNA/μg of total DNA.

### 2.6. Analysis of Expression of Plant Defense-related Genes

For plant gene expression analysis, aliquots of 25 mg of lyophilized wheat leaves sampled as indicated in [6] were ground in liquid nitrogen and treated for RNA extraction. RNA extraction was performed with the TRI reagent method (Sigma–Aldrich) according to manufacturer’s instructions; cDNA was obtained using the first strand cDNA synthesis Super Script II RT-PCR kit (Thermo Fisher Scientific, Waltham, MA, USA). RT-PCR was performed as described [28]. To identify target genes and primer (Appendix A) homologs in wheat, we used the NCBI, DFCI Wheat Gene Index, an International Wheat Genome Sequencing Consortium chromosome survey sequence repository at Unité de Recherche Génomique Info INRA databases, and specific references (for the homologues of *cerk1* see [29]; for the homologue of *mpk3* see [30]). The β-tubulin gene of *T. turgidum* subsp. *durum* (acc. N. AJ971820.1; for 5’–GCTGCTGTATTGCAGTTGGC–3’; rev 5’–AAGGAATCCCTGCAGACCAG–3’) was employed as a housekeeping reference to normalize the expression of the target genes in the wheat leaves. Previous trials showed that Tramesan was effective in limiting the growth of pathogens and in reducing the symptoms on leaves. Ruling out a direct antimycotic effect of Tramesan (M. Reverberi, unpublished results) on *P. nodorum* ST15465 and *Z. tritici* ST18258, we carried out experimental tests to evaluate whether this exo-polysaccharide was able to induce defense responses in wheat plants. In this regard we analyzed the expression of some marker genes of defense response in the durum wheat variety Svevo: the pathogenesis related *PR1, PR4, PR9, CHIT2* (coding for a chitinase active against fungi)*, MCA2* (coding for a metacaspase)*, NADPH-ox* (coding for the anion superoxide forming NADPH oxidase)*, PAL* (coding for the phenylalanine-amonialyase), *CERK1* (coding for the chitin related receptor CERK), and *MPK3* (coding for the MAP kinase MPK3). Indeed, this variety was more susceptible to both pathogens.

The relative expression of wheat genes was calculated following the 2^−ΔΔCt^ method using untreated samples as calibrators. RT-PCR was prepared in a 20 μL reaction mixture containing SYBR green JumpStart Taq Ready Mix 1X (Sigma–Aldrich), 3 mM MgCl_2_, and 0.5 mM β-tubulin. RT-PCR was performed in a LineGene K PCR detection system (Bioer, Hangzhou, China). *PR1, PR4, PR9, CHIT2, MCA2, NADPH-ox, PAL*, *CERK1*, and *MPK3* expressions were evaluated at inoculation time (t_0_) and at 24 hours after inoculation (hai) on Svevo artificially infected with: (1) *P. nodorum* not treated (Inf Pn) or treated (Tr + Inf Pn) with 3.3 μM of Tramesan or (2) *Z. tritici* not treated (Inf Zt) or treated (Tr + Inf Zt) with 3.3 μM of Tramesan.

### 2.7. Plant Hormones Analysis 

Salicylic acid (SA) and jasmonic acid (JA) were sampled as indicated in Oliver et al., 2012 [6] and extracted from wheat leaves in all the experimental conditions described above. The quantification was done by the addition of the internal standard 1-naphthaleneacetic acid (NAA, MW 186.21 g mol^−1^), at 5 μM final concentration. An amount of 30 mg of lyophilized durum wheat leaves was ground with liquid nitrogen, mortar, and pestle. Hormones were extracted with 750 μL of MeOH:H_2_O:HOAc (90:9:1, v/v/v), mixed, and centrifuged for 1 min at 10,000 rpm. The supernatant was collected, and the extraction was repeated. Pooled supernatants were dried under nitrogen gas flux. The dried samples were resuspended in 200 μL of 0.05% HOAc in H_2_O-acetonitrile (85:15, v/v). The analysis of SA and JA was performed with a LC–MS/MS Agilent 6420. The acquisition was in MRM negative ion mode [M-H]^−^. Chromatographic separation was performed with a Zorbax ECLIPSE XDB-C18 rapid resolution HT 4.6 × 50 mm 1.8 μm p.s. column (Agilent Technologies, Santa Clara, CA, USA) at room temperature, and the injected volume was 10 μL. The mobile phases consisted of A: H_2_O containing 0.05% HOAc, and B: acetonitrile. The elution gradient was as follows: 0–3 min 15% B, 3–5 min 100% B, 5–6 min 100% B, 6–7 min 15% B, 7–8 min 15% B. The gradient was followed by 5 min for re-equilibration. The flow rate was constant at 0.6 mL·min^−1^. The injector needle was washed with the mobile phase in the wash at the end of each HPLC run. Nitrogen was used as the nebulizing and desolvation gas. The temperature and flow of the drying gas were 350 °C and 10 mL·min^−1^, respectively, with a nebulization pressure of 20 psi. The capillary and cone voltage were 4000 V. The main transition and qualifier ions were 137.2🡪92.9, 137.2🡪64.8 for SA (CE 20, FV 135), 209.2🡪59.1, 209.2🡪41.3 for JA (CE 28 FV 135), and 245🡪180.8 for the internal standard NAA (CE 16 and FV 100). To quantify hormone levels, a standard curve was generated for both compounds in the 0.01 μM to 10 μM range. Trend curves were linear within this range and the equations correlating relative (area) abundance and molarity were y = 3432.2x, R^2^ = 0.9973 for SA and y = 541.22x, R^2^ = 0.9813 for JA, respectively.

### 2.8. Statistics

Data were analyzed by t-test, ANOVA and, in case of statistical significance, Fisher’s multiple comparison tests between means were performed (XLSTAT, Addinsoft, Paris, France).

## 3. Results

### 3.1. Phytotron Trials 

The SNB and STB severity on the leaves of two commercial varieties of durum wheat, Svevo and Duilio, was calculated based on Liu’s scale, a 0-to-5 qualitative lesion-type rating scale that assesses symptoms on leaves at 7 dai (Table 2 and Figure 1). Phytotron trials were performed following the same timeline for both pathogens notwithstanding the fact that these are not the correct and proper ways to test pathogenicity of *Z. tritici.*

Results showed that plants treated with Tramesan had fewer symptoms compared to the untreated ones. Svevo was more susceptible to the disease (mainly *P. nodorum*) than Duilio. By quantifying the fungal DNA within leaf tissues at 0, 3, and 9 dai (Figure 1), we found that the pathogen load dramatically decreased under Tramesan treatment by up to 99% for *P. nodorum*, and up to 75% for *Z. tritici*.

### 3.2. Field Trials 

The plants treated with Tramesan showed a lower severity of disease (Table 3; Appendix A and Appendix A), even if results were less intense than those under controlled growth conditions (symptoms decreased by up to 25% for *P. nodorum* and up to 30% for *Z. tritici*). In addition, Tramesan treatment did not affect grain yields (5.29 ± 0.05 t/ha) compared to untreated plots (5.28 ± 0.08 t/ha) that only displayed limited SLBC leaf infection in our assay.

### 3.3. Expression of Defense Genes and Hormones in Durum Wheat Leaves

Leaves were harvested at 48 h after treatment. RT-PCR showed a significant difference in gene expression in plants treated with Tramesan; notably, the enhancement of the expression of genes involved in the onset of plant defenses [19] occurred at 48 h after treatment compared to untreated plants. This significant difference of expression between Tramesan-treated and untreated plants was found in almost all the defense genes analyzed (Figure 2; Appendix A).

The assay of SA and JA levels in infected leaves provided straightforward results (Figure 3; Appendix A). Notably, JA was induced at an approximately 3-fold higher concentration at 48 h after Tramesan application on plants. SA was down-modulated both in the control (mock) and treated (T) plants, the latter registering a lower decrease in SA compared to the mock plants.

Recognition and signal transduction-related *cerk1* and *mpk3* expressions were triggered by fungal infection and were even more enhanced in plants treated with Tramesan (Figure 4). Fungal infections depressed both SA and JA production at 24 hai (fold change (FC) in infected vs. non-infected samples: 0.15 for SA and 0.4 for JA). Pre-infection treatment with Tramesan (Tr + Inf) had little effect on SA (FC of 0.25 for Tr + Inf vs. non-infected control) but triggered JA biosynthesis (FC of 6.2 for Tr + Inf vs. non-infected control) (Appendix A).

Potential indicators of plant response to damage, phenylalanine ammonia lyase *PAL* that is the main hub for the phenylpropanoid pathway and metacaspase-encoding *MCA2* that is active in the programmed cell death induced by biotic stress, were expressed at high levels even in the absence of the pathogen; while *PAL* did not change its expression profile after fungal infection (even though Tramesan strongly elicited its expression upon fungal infection), *MCA2* was significantly upregulated after fungal inoculation (Appendix A). *P. nodorum* infection down-regulated the expression of the respiratory burst oxidase (*RBOH*); the product of this gene represents an indicator of stress response in plants. Similarly, *Z. tritici* infection pushed down *RBOH* expression; only in this latter case did Tramesan aid its stimulation (Figure 4). This could be explained by the set of genes under the putative control of SA and JA, even when different outcomes were observed between the two pathogens. In fact, under *P. nodorum* infection, the expression of the SA-dependent pathway *PR1* and *PR9* decreased in both treatments (Inf Pn and TR + Inf Pn), whereas the JA-modulated *PR4* and *Chit2* showed an opposite trend: unaffected or slightly down-regulated in Inf Pn and strongly up-regulated in Tr + Inf Pn (Figure 4 and Appendix A). On the contrary, Tramesan, in the presence of *Z. tritici*, up-regulated SA-dependent genes (e.g., *PR1*), whereas JA-controlled genes (e.g., *PR4*) were almost unaffected (Figure 4 and Appendix A).

## 4. Discussion

Septoria Disease Complex is a major issue for European wheat farmers. Excessive cost of fungicides and the emerging fungicide resistance of pathogens are consistent threats to wheat yield. Even though several loci for resistance to SLBC were identified so far, a deficit in availability of resistant lines still remains [31]. One challenge in developing stable resistant lines might lie in the complexity of the SLBC disease. The two main effectors of this disease, *Z. tritici* and *P. nodorum*, play their pathogenic role in wheat differently. These differences emerge in the initial stage of infection: the hemibiotrophic *Z. tritici* avoids plant recognition by camouflaging its PAMP (e.g., chitin with Mg3LysM); the necrotrophic *P. nodorum* induces plant cell death via direct effects (e.g., ToxA vs. Tsn1 sensitivity gene) [10]. Nonetheless, twenty major *Stb* genes providing a qualitative resistance to STB have been identified so far [32]. Therefore, it is reasonable to suppose that novel SLBC-resistant lines could be on the market within few years. Nowadays, the SLBC is becoming more and more of a constraint for wheat production worldwide since the emergence of *Z. tritici* and *P. nodorum* strains QI and SDHI-resistant [31].

The more and more restrictive country-based laws (e.g., EC/128/2009) confine the ability to eradicate the problem; moreover, in Europe, today the highest costs in wheat cultivation are those related to fungicides against SLBC. In view of this, the search for alternative products in limiting SLBC in wheat is gaining momentum [29,30]. In these studies, diverse natural elicitors (e.g., natural polysaccharides: λ-carragenan) can significantly limit SLBC in wheat (i.e., up to 70% under controlled growth conditions) by enhancing plant defenses exploiting both jasmonates and salicylate signaling pathways.

We show that Tramesan—a basidiomycete-derived exo-polysaccharide—could represent a natural, user-friendly, low-cost alternative to protect crops. Specifically, our results indicate that Tramesan supports wheat plants in controlling SLBC causal agents: *P. nodorum* and *Z. tritici.* However, Tramesan’s mode of action is still an outstanding question.

We report that Tramesan is effective at augmenting the synthesis of defense-related hormones, notably JA, and of some markers of plant defenses (*PR1* and *PR4 inter alia*). This effect occurs within 48 hours after Tramesan is applied, albeit not all the markers tested react linearly: the mitogen-activated protein MPK3 and the LysM receptor kinase CERK1 display high basal levels [31,32] even if induced by Tramesan and fungal infection. In contrast, *PAL*, *MCA2*, and *RBOH* are evidently controlled by additional inputs other than stress (ABA (–TAS14) and auxin (–PIN2)) [33,34]. The set of genes putatively regulated by SA, such as *PR1* and *PR9*, or by JA, such as *PR4* [35,36], showed a straight correlation with the hormone levels. In the case of SA-controlled genes, NPR1 is the most likely mediator of the stimulus [37,38,39,40,41].

Based on our data, Tramesan elicits a high basal activation of plant defenses, even though the “nature” of this elicitation process remains to be determined. In this perspective, Tramesan can be envisioned as a MAMP that plants recognize as a non-self biomolecule to induce defense. The search for compounds, mainly microbe-derived, able to efficiently induce plant defenses is gaining momentum in the process of elucidating induced systemic resistance and defense priming patterns [14,16]. Our results did not allow assessment of whether Tramesan could induce an immune response (IR) or prime plant defenses; further studies are needed in this regard. Data could suggest that Tramesan is more than a “simple” MAMP: the onset of defense appears to be modulated over time, triggered by subsequent infection and effective at reducing pathogen growth and disease symptoms. Is it costly? Probably not: yield data from field experiments suggest that Tramesan did not affect durum wheat production. We hypothesize that Tramesan is recognized by specific receptors that, in turn, activate pathways leading to an antioxidant response in the host. Tramesan acts in several organisms (mammals, fungi, and plants) as an activator of antioxidant defenses [20]; when employed against mycotoxigenic fungi, it switches off aflatoxin synthesis—a process closely related to the antioxidant system of the cell—in *Aspergillus* section Flavi [42]. Therefore, along with inducing plant defenses, Tramesan can lower circulating reactive oxygen species (ROS) at the plant–pathogen interface, thus creating a less favorable environment for the necrotrophic growth of plant pathogens such as *P. nodorum.* Several receptors in various organisms are known to act as ligands of fungal glucans, and to trigger various responses (e.g., innate immunity to cell death or symbiosis [42,43,44]). Most recently, Gubaeva and colleagues proposed the “slipped sandwich” model for explaining the ability of AtCERK1 in bound and transduced oligochitosan (the most effective with a degree of polymerization up to 7-DP7) to induce defense reactions in *A. thaliana* [44]. Similar results were found in common wheat with the identification of receptor kinase proteins putatively able to bind oligochitosan [45].

In this context, we propose that Tramesan acts as a ligand for still unknown inter-kingdom conserved receptors, and that this recognition elicits host defenses to limit pathogen development and disease. The ability that Tramesan demonstrated in modulating cell antioxidation in other systems, along with the well-established relationship between plant defense onset and antioxidant modulation [45], suggest that the triggering of defense-related hormones and genes occurs at the very early steps following pathogen recognition.

Further studies in the field, under natural infection conditions, should allow us to better assess if the priming and memory effect of Tramesan in host plants does actually occur, and also its real effectiveness in controlling SLBC.

## Figures and Tables

**Figure 1 biomolecules-10-00608-f001:**
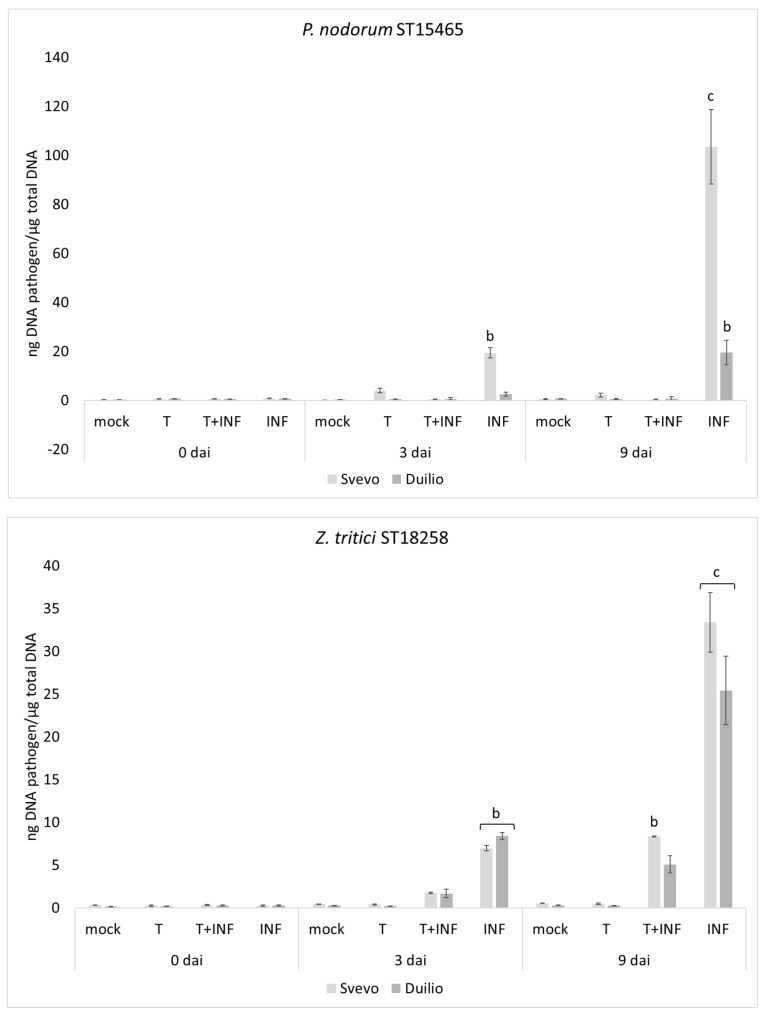
*P. nodorum* ST15465 (**upper**) and *Z. tritici* ST18258 (**lower**) growth assessed by qPCR. At 0, 3, and 9 days after inoculation (dai) in durum wheat (vars. Svevo and Duilio). Mock: without Tramesan, non-inoculated. T: treated with Tramesan; T + Inf: treated with Tramesan, inoculated with the pathogen; Inf: inoculated with the pathogen. Values represent the mean ± SE as described in the methods section. Small capital letters in the chart represent the significantly different groups (*p* < 0.05; Fisher’s test).

**Figure 2 biomolecules-10-00608-f002:**
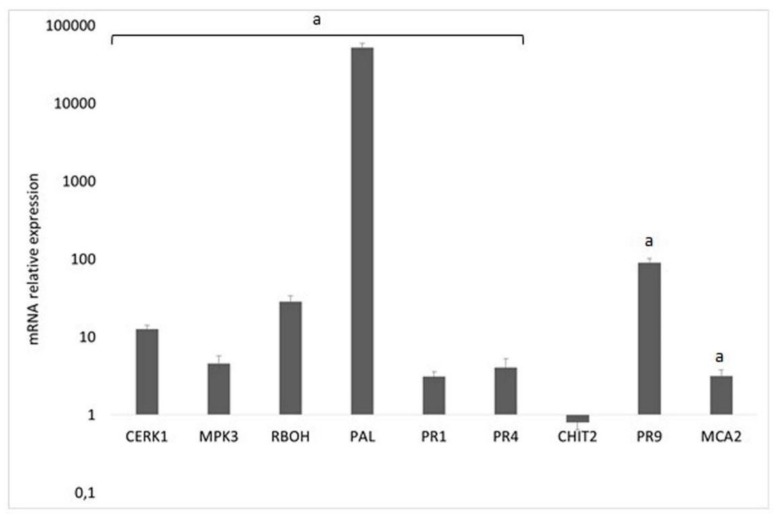
Expression profile of transcripts of genes *CERK1, MPK3, NADPHox (RBOH), PAL, PR1, PR4, CHIT2, PR9, and MCA2* in durum wheat var. Svevo at 48 h after treatment with Tramesan (3.3 μM). Expression is relative to values in untreated plants (control) and values represent the mean ± SE as described in the Methods section. Small capital letters in the chart represent the significantly different groups (*p* < 0.05; Fisher’s test).

**Figure 3 biomolecules-10-00608-f003:**
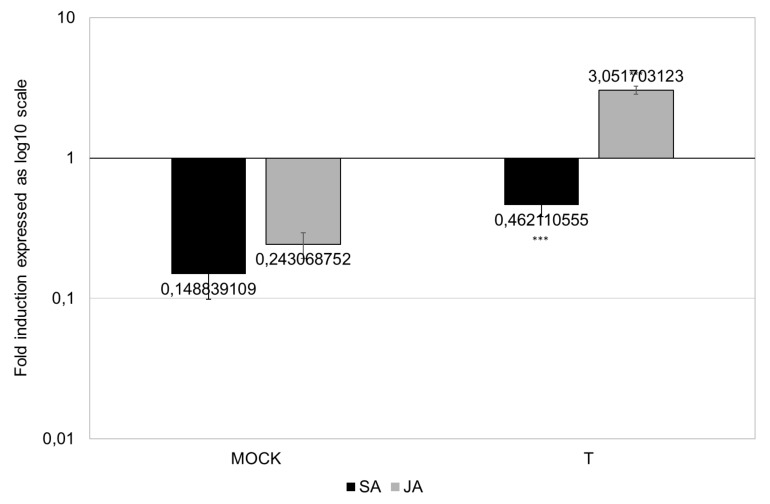
Fold induction (reported in log_10_ scale) of salicylic acid (SA) and jasmonic (JA) acid in durum wheat var. Svevo, after 48 h treatment (T) or untreated (mock) with Tramesan (3.3 μM). Values represent the mean ± SE as described in the methods section. Small capital letters in the chart represent the significantly different groups (*** *p* < 0.001; t-test).

**Figure 4 biomolecules-10-00608-f004:**
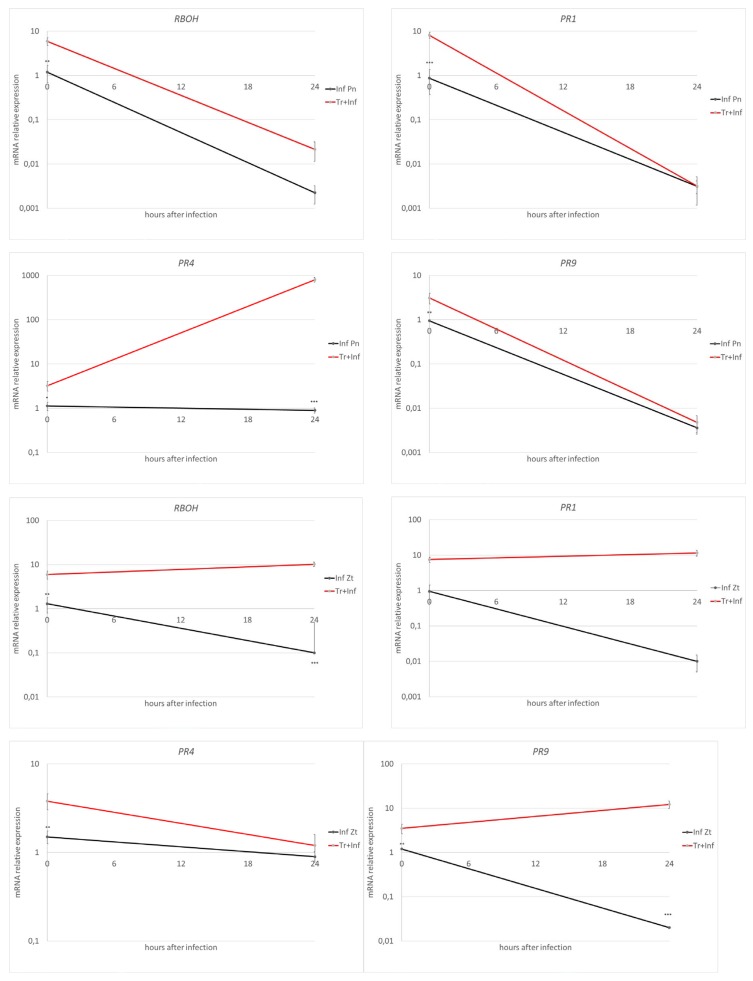
Expression profile of transcripts at 0 and 24 hours after infection (hai) of genes *RBOH, PR1*, *PR4*, and *PR9* in durum wheat var. Svevo infected with *P. nodorum* ST15465 (Inf Pn) or *Z. tritici* ST18258 (Inf Zt) and infected with pathogens and pre-treated (−48 hai) with Tramesan (Tr + Inf). Expression is relative to values in untreated plants (CT−; without Tramesan, non-inoculated) and values represent the mean ± SE as described in the methods section. Asterisks represent the significantly different values (* *p* < 0.05; ** *p* < 0.01; *** *p* < 0.001; t-test).

**Table 1 biomolecules-10-00608-t001:** Different experimental conditions used on durum wheat cultivars to study the effect of Tramesan on StagonosporaNodorumBlotch and SeptoriaTriticiBlotch in Montelibretti (Rome) and in Rome (Italy) field trials performed by the Council for Agricultural Research and Economics—Research Centre for Engineering and Agro-Food processing (CREA-IT) in the 2014–2015 growing season.

Treatment	Application Time (GS)	Active Ingredient	Dose (mL/m^2^)
Untreated control (non-inoculated and non-treated)	--	Water	40
T	47	Tramesan	40
Inoculated with *P. nodorum* ST15465	49	Water	40
T and inoculated with *P. nodorum* ST15465	47; 49	Tramesan	40
Untreated control (non-inoculated and non-treated)	37	Water	40
T	37	Tramesan	40
Inoculated with *Z. tritici* ST18258	39	Water	40
T and inoculated with *Z. tritici* ST18258	37; GS 39	Tramesan	40

**Table 2 biomolecules-10-00608-t002:** Disease severity as measured at 7 dai at the second leaf stage according to Liu’s scale (0–5) [23] in the durum wheat varieties (Svevo and Duilio) inoculated with *P. nodorum* ST15465 or *Z. tritici* ST18258. Mock: without Tramesan, non-inoculated. T: treated with Tramesan; Tr + Inf: treated with Tramesan, inoculated with the pathogen; Inf: inoculated with the pathogen. Fisher test on *n* = 48 × 2 biological repetitions.

	*Durum Wheat*			
	*P. nodorum* ST15465		*Z. tritici* ST18258	
	Svevo	Duilio	Svevo	Duilio
**Mock**	0^a^	0^a^	0^a^	0^a^
**T**	0^a^	0_a_	0^a^	0^a^
**Tr + Inf**	1^ab^	1^ab^	0.7^ab^	0.7^ab^
**Inf**	4^b^	2^b^	2^b^	1.7^b^
***F test (p value)***	*2.5 × 10^−5^*	*0.031*	*0.0015*	*0.025*

^a,b^ refer to Figure 1 upper and lower.

**Table 3 biomolecules-10-00608-t003:** SNB and STB severity on flag leaf (%) in durum wheat (var. Svevo) at growth stage GS 83 grown in Montelibretti (Rome) and in Rome fields in the 2014–2015 growing season. Mock: without Tramesan, non-inoculated. T: treated with Tramesan; T + Inf: treated with Tramesan, inoculated with the pathogen (SNB: *P. nodorum* ST15465; STB: *Z. tritici* ST18258); Inf: inoculated with the pathogen. Values represent the mean ± SE of three technical repetitions (*n* = 3).

	Svevo	
	Severity on Flag Leaf (%) SNB	Severity on Flag Leaf (%) STB
**Mock**	3.2^ab^	1.2^a^
**T**	0.8^a^	0.5^a^
**Tr + Inf**	28.5^b^	32.4^b^
**Inf**	36.5^c^	47.5^c^
***F test (p value)***	*4.65 × 10^−4^*	*1.73 × 10^−5^*

Means followed by different letters are significantly different (the level of significance p is shown in the table). ^a,b,c^ refer to Figure 1 upper and lower.

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
