# Peer review of "Tramesan Elicits Durum Wheat Defense against the Septoria Disease Complex"

_biomolecules, 2020, doi:10.3390/biom10040608_

Round 1

Reviewer 1 Report

The manuscript “Tramesan elicits durum wheat defence against the Septoria Leaf Blotch Complex” by V. VALERIA et al. describes the protective effect of a fungal polysaccharide on wheat leaf blotch fungal diseases. Tramesan is a fungal polysaccharide recently identified by Authors that is secreted by the fungus Trametes versicolor, and was shown by Authors to inhibit human tumor cell growth and fungal mycotoxin production, as well as inducing oxidative stress in fungal and human cells.

In this follow-up paper, Authors have tested the effect of Tramesan on fungal wheat diseases by treating wheat plants with Tramesan before infection either in controlled conditions or in field tests, using two fungal pathogens (Parastagonosora nodorum, and Zymoseptoria tritici). Preliminary results on the protective effect of Tramesan on P. nodorum wheat infection were described in another paper (PLoS ONE 2017. 12(8): e0171412.). However, this novel study is much more complete and convincing than this first report. Indeed, Authors clearly showed that Tramesan has a strong protective effect on P. nodorum wheat infection in controlled conditions and a significant effect in field tests, despite it does not inhibit P. nodorum mycelial in vitro growth, as described previously.

The protective effect of Tramesan is less obvious for Z. tritici. Indeed, Authors have not used a correct experimental setting for this fungal disease in controlled conditions, as they have stopped the experiment 9 days after inoculation. This is not sufficient for the development of Z. tritici symptoms that occurs only after 10-12 days. Usually, symptoms are recorded 21 days after inoculation, as the % of leaf area covered with pycnids (see Karisto et al. Phytopathology. 2018. 108(5):568-581, and Rudd et al. Plant Physiol. 2015 167(3):1158-85). Therefore, the Z. tritici disease severity assessment of Authors in Table 1 is not appropriate. Still, Authors could use their data from Figure 1, in which they have quantified Z. tritici in infected wheat leaves, as it showed some inhibition of fungal development. However, Authors have to state that they have not use the correct experimental design for this disease. The field tests for Z. tritici have a correct design since Authors have followed up the disease for a sufficiently long period after inoculation, and they have observed a significant reduction in disease development after Tramesan treatment.

Authors have also tried to decipher how Tramesan protects wheat from fungal diseases. Their hypothesis is that this compound has no direct toxic effect on fungal growth, as shown for P. nodorum in a previous paper, but not for Z. tritici, but induces plant defenses. They have design experiments to follow up the induction of wheat defense genes after Tramesan treatment with or without infection. They have shown that Tramesan alone induces the expression of wheat genes known to be involved in defense to fungal diseases (Figure 2). Results from salicylic and jasmonate assays are not really convincing, nor easy to interpret. Experiments involving the combination of Tramesan treatments and infection (Figure 4) are not easy to follow with the actual settings. Authors should display the expression during infection +/- Tramesan of each gene as a kinetics (to, 24h). This would allow following up more easily the expression profile of a given gene during this experiment. One question that Authors should answer is what is the fold increase or decrease in the expression of a given defense gene due to Tramesan either at to or 24 after infection. It is difficult to know what is happening with actual Figure 4.

Overall, the description of the protective effect of Tramesan against wheat fungal diseases as a possible plant defense priming compound is novel and of interest. In particular, its effect in field conditions (-30% disease) is encouraging, since many plant defense priming compounds have only a low protective effect in field conditions.

However, I think that this manuscript is not ready for publication, and requires significant improvements. Authors have either to take off Z. tritici experiments in controlled conditions (keeping field test), or explain why they have only taken into account early stages of Z. tritici infection. Authors have to re-organize their assay of wheat defense induction by Tramesan in a more concise and demonstrative way (see before). Authors have also to compare their results to those obtained on similar wheat diseases with other priming compounds/elicitors (not quoted in their manuscript), since this is a quite active field (see for example: Ors Met al. Phytopathology. 2019 109(12):2033-2045 ; Le Mire G et al. Phytopathology. 2019 109(3):409-417). Using Bion as a positive control and a fungicide as a negative control would have been beneficial for their demonstration. Authors have also to improve their manuscript both for its English and for its organization. For example, I would recommend to place all results Figures/Tables in the result section (ie move them from the material and methods section), and comment them in more detail than the few actual sentences (8 lines only). I would also recommend placing the field test just after the phytotron assays. Then Authors could describe their plant defense experiments, but in a more concise way (some part of Figures could be moved as sup files). Authors have also to provide more focused introduction and discussion sections, as some existing topics are not too relevant for the manuscript. A more exhaustive description of Tramesan biological activities including those observed in fungi, would be helpful for the reader in the introduction.

Author Response

Thanks a lot for your kind and precious suggestions. We try to do our best to answer your questions. Please find the answers in the pdf uploaded

Reviewer 2 Report

Abstract

Page 1: line 19 -20. Rephrase   “adversely affects human and animal health and the environment”.

Page 1: line 26-27. Report p values to show level of significance.

Page 1: line 30-31. Report yield values.

Introduction:

Provide introduction to Tramesan, its successful treatments and applications

Material and Methods:

General comments:

You have selected moderately susceptible varieties? Can you please provide their previous individual disease ratings? Why not select a susceptible variety as a control? How old were the fungus (septoria and nodorum) isolates. As longer storage can affect infection ability. Also, were the isolates tested for infection prior to main experiments, to see if viable? The Septoria has two stage infection processes- asymptomatic and symptomatic. You scored diseased plants after 9 dai, why? Generally septoria symptoms start to appear after 12 dai and are finally scored at 21 dai. Were the plants scored at 21 Dai? Briefly describe the Liu’s rating scale. Liu’s rating scale was developed for Stagonospora nodorum Leaf Blotch in Wheat? How well it encompassed the septoria disease ratings as it normally takes 21 dai for complete comparative symptoms assessment. Is there any information on the effect of Tramesan on pycnidia formation of septoria? Which is very important to reduce the inoculum load and number of cycles during the field season. Please provide information about different time points selected for gene expression analysis. How you selected the defence related genes, please provide full name of the gene, selection methodology and cite the relevant studies? Salicylic acid (SA) and jasmonic acid (JA) were extracted from wheat leaves in all the experimental conditions described above. Please clarify how sampling and processing was done, in terms of varieties and pathogens. Were the glasshouse experiment repeated? Again, in field trials, are there any disease standards used. These standards serve baseline for level of disease epidemic in the field. Since you assessed disease after 7-10 days until ripening. Can you calculate relative Area under disease progress curve? To visualize how the disease progressed overtime.  

Page 4: line 190-191. Should be moved to results or discussion section.

Results:

General comments:

Results need to be clearly written and under separate headings as in the material and methods section. Raw phenotypic data for seedling and adult plant field trial will be beneficial. Please add as supplementary file. Please provide leaf response photo of all experimental condition to visualise the Tramesan effect on disease. Leaves were harvested at 48 hours after treatment. Can you please double check? Statements like “The expression of these genes could be indicative of a defence priming status of the plant, as 286 elsewhere suggested [20]” should be moved to discussion section. What was hormone fold induction for Duilio? Can you explain why did not observed a Yield advantage when treated with Tramesan?

Figure 1:

Can you explain why Svevo at 3dai treated with Tramesan has higher pathogen DNA than 9 dai. Amount of DNA is comparative to some of the infected treatments. Can you please modify graphs to see the lower error bar?

Figure 2:

Can you explain why Svevo at 3dai treated with Tramesan has higher pathogen DNA than 9 dai. Amount of DNA is comparative to some of the infected treatments.

Author Response

(The authors gave the same response as above.)

Round 2

Reviewer 1 Report

Authors have made significant improvements, and I think that the manuscript is almost ready for publication. Still some minor modifications are required (see below). One comment concerning the timing of symptoms development for Z. tritici. It is now recognized that the transient chlorosis observed during Z. tritici early leaf infection is not reliable, as it is too variable, not quantitative, and not necessarily related to the outcome (necrosis and pycnidia formation).

Abstract

Despite clear improvements in their manuscript, Authors still have to be more cautions with their Abstract

- Foliar-spraying of Tramesan (3.3 μM) in SLBC-susceptible varieties of durum significantly diminished symptoms of Stagonospora Nodorum Blotch (SNB) and Septoria Tritici Blotch (STB) by 75% and 65%, respectively. Reviewer comment: Decrease disease incidence is closer to Authors findings, since symptoms at 7 days in controlled conditions is not appropriate for Z tritici. For controlled conditions, I think that it is better to use results obtained by quantifying fungal growth using qPCR than symptoms. Indeed, decrease in symptoms is only highly significant for the combination Svevo x SNB, and inappropriate for STB for the timing used. One could think of a sentence like: Foliar-spray of Tramesan (3.3 μM) on SLBC-susceptible wheat Durum cultivars before inoculation of Stagonospora Nodorum Blotch (SNB) and Septoria Tritici Blotch (STB) significantly decrease disease incidence both in controlled conditions (SNB: - 99%, STB-75%) and field assays (SNB: -25%, STB: -30%).

- We show that Tramesan elicits wheat defence against SNB and STB by augmenting the synthesis of defence-related hormones, notably JA and SA, Reviewer comment: Not true for SA. For JA, Authors do not know if there is an increase in synthesis or a decrease in degradation (both are possible). Better to state: We showed that Tramesan increased the levels of JA, a plant defense-related hormone.

- that in turn switch on the expression of defense markers of defence (PR1 and, PR4. inter alia). Reviewer comment: Authors cannot state that increased JA levels induced the expression of these defense genes (not the scope of this paper). Wiser to state that: Tramesan also increased the early expression (24 hpi) of plant defense genes such as PR4 for STB infected plants, and RBOH, PR1 and PR9 for STB. These results suggest that Tramesan protect wheat by eliciting plant defenses, since it has no direct fungicidal activity (if true).

- In field experiments, the yield of durum wheat plants treated with Tramesan was similar to that of untreated ones. Reviewer comment: better to state In field experiments, the yield of durum wheat plants treated with Tramesan was similar to healthy untreated plots.

Introduction

- Similarly, Tramesan induced enhancement of antioxidant response in wheat leaf cells provides better resistance to some fungal pathogens. Reviewer comment: delete induced , Add the reference

- we posit Reviewer comment: better to use we propose

Material and methods

- Zymoseptoria tritici, strain 18258, was isolated at CREA-IT and stored at -20 °C. Reviewer comment: Is this strain isolated from durum wheat? Please add this important information to material and method. Is this train pathogenic on durum wheat in controlled assays (with 3 weeks delay after inoculation)? Please add this important information to material and method. Is this strain pathogenic on bread wheat in controlled inocultation. Please add this important information to material and method. Indeed, Zt strains could be either pathogenic to both wheat types or only one type.

- Kernels of two Italian commercial varieties of durum wheat (Svevo and Duilio; Syngenta Italia and SIS società italiana sementi, respectively, ) of durum wheat (T. turgidum subsp. durum (Desf. Husn.), moderately susceptible to the SLBC, Reviewer comment: What is the level of susceptibility of these two Durum wheat cultivars to P nodorum on one side and Z tritici on the other side? It should be quite different, since resistance/susceptibility to these diseases are controlled by very different genetic factors. For example, which susceptibility genes to P nodorum are found in these cultivars, if known?

- Each plot in the two fields contained n = 250 plants (plants were fewer than seed probably for the considerable presence of birds at the time of sowing). The trials were performed in three plots, each containing all treatments per thesis (mock, T, Inf, T+Inf). Inoculation for the Artificial contamination in the field plots was prepared as indicated in “Preparation of fungal inoculum.” Reviewer comment: It seems that the same P nodorum and Z tritici strains have been used for Phytotron and Field trials, but this is not clearly stated. Please add the name of the strain used in all the Figure , Tables, so there is no doubt about it.

Result Section

- Table 2. Disease severity at the second leaf stage according to Liu’s scale (0-5) in the durum wheat varieties (Svevo and Duilio) inoculated with P. nodorum or Z. tritici. Mock: without Tramesan, non-inoculated. T: treated with Tramesan; Tr+Inf: treated with Tramesan, inoculated with the pathogen; Inf: inoculated with the pathogen. Fisher test on n=48 × 2 biological repetitions. Reviewer comment: Add the time at which symptoms were scored, ie 7 days, and give the reference of the scale used in legend.

- Comments on Table 2

Results showed that plants treated with Tramesan had fewer symptoms compared to the untreated ones, and that Tramesan reduces disease severity by P. nodorum (up to 75%) more than that by Z. tritici (up to 65%), even if these differences are not statistically significant (p=0.06). Reviewer comment: change, see abstract

- Comments on Figure 1

By quantifying the fungal DNA within leaf tissues at 0, 3, and 9 dai (Fig. 1 a,b), we found that the pathogen load dramatically decreased under Tramesan treatment by up to 99% for P. nodorum, and up to 75% for Z. tritici. Reviewer comment: Please give the statistical significance of the comparison in leaf fungal DNA content between Inf vs Inf+T for each cultivar x strain combination. This is not clear actually in Figure 1, as it looks like Authors have compared the two diseases for infection on one side and infection + T on the other side. Authors could use a specific table if it is easier than statistics labels on Figure 1, as these data are very important for demonstrating that Tramesan strongly reduces infection, as assayed by quantifying fungal growth.

- Figure 1 Reviewer comment: It would be better to group together the kinetic of infection for each strain/cultivar pair, since it is quite different quantitatively for each interaction (really not the same scale). This would make Figure 1 more easy to follow.

- Furthermore, we analyzed the yield of the harvested grain, proving that the Tramesan treatment (5.29 ± 0.05 t/ha) did not affect it compared to the untreated plots (5.28 ± 0.08 t/ha). Reviewer comment: better to say: In addition, Tramesan treatment did not affect grain yields (5.29 ± 0.05 t/ha) compared to untreated plots (5.28 ± 0.08 t/ha) that only displayed limited SLBC leaf infection in our assay.

- The measurement of the content of defense-related hormones such as SA and JA….Reviewer comment: Better to say The assay of SA and JA levels in infected leaves …

- Figure 3: Reviewer comment: it would be nice to add value for SA and JA relative abundance just under the each box.

- Figure 4 has been really improved. Reviewer comment: It could be nice to use colored lines (Red, Blue) to highlight infection + T compared to infection (black).

Discussion

- we posit Reviewer comment: better to use we propose

Author Response

Authors have made significant improvements, and I think that the manuscript is almost ready for publication. Still some minor modifications are required (see below). One comment concerning the timing of symptoms development for Z. tritici. It is now recognized that the transient chlorosis observed during Z. tritici early leaf infection is not reliable, as it is too variable, not quantitative, and not necessarily related to the outcome (necrosis and pycnidia formation).

Thanks for your kind suggestions. We agree with your additional comments and we modified accordingly the text and tried to answer your doubts, where possible.

Abstract

Despite clear improvements in their manuscript, Authors still have to be more cautions with their Abstract

- Foliar-spraying of Tramesan (3.3 μM) in SLBC-susceptible varieties of durum significantly diminished symptoms of Stagonospora Nodorum Blotch (SNB) and Septoria Tritici Blotch (STB) by 75% and 65%, respectively. Reviewer comment: Decrease disease incidence is closer to Authors findings, since symptoms at 7 days in controlled conditions is not appropriate for Z tritici. For controlled conditions, I think that it is better to use results obtained by quantifying fungal growth using qPCR than symptoms. Indeed, decrease in symptoms is only highly significant for the combination Svevo x SNB, and inappropriate for STB for the timing used. One could think of a sentence like: Foliar-spray of Tramesan (3.3 μM) on SLBC-susceptible wheat Durum cultivars before inoculation of Stagonospora Nodorum Blotch (SNB) and Septoria Tritici Blotch (STB) significantly decrease disease incidence both in controlled conditions (SNB: - 99%, STB-75%) and field assays (SNB: -25%, STB: -30%).

- We show that Tramesan elicits wheat defence against SNB and STB by augmenting the synthesis of defence-related hormones, notably JA and SA, Reviewer comment: Not true for SA. For JA, Authors do not know if there is an increase in synthesis or a decrease in degradation (both are possible). Better to state: We showed that Tramesan increased the levels of JA, a plant defense-related hormone.

- that in turn switch on the expression of defense markers of defence (PR1 and, PR4. inter alia). Reviewer comment: Authors cannot state that increased JA levels induced the expression of these defense genes (not the scope of this paper). Wiser to state that: Tramesan also increased the early expression (24 hpi) of plant defense genes such as PR4 for STB infected plants, and RBOH, PR1 and PR9 for STB. These results suggest that Tramesan protect wheat by eliciting plant defenses, since it has no direct fungicidal activity (if true).

- In field experiments, the yield of durum wheat plants treated with Tramesan was similar to that of untreated ones. Reviewer comment: better to state In field experiments, the yield of durum wheat plants treated with Tramesan was similar to healthy untreated plots.

We have modified (under revision mode) the abstract according to your suggestions.

Introduction

- Similarly, Tramesan induced enhancement of antioxidant response in wheat leaf cells provides better resistance to some fungal pathogens. Reviewer comment: delete induced , Add the reference

- we posit Reviewer comment: better to use we propose

We have modified (under revision mode) the sentences according to your kind suggestions.

Material and methods

- Zymoseptoria tritici, strain 18258, was isolated at CREA-IT and stored at -20 °C. Reviewer comment: Is this strain isolated from durum wheat? Please add this important information to material and method. Is this train pathogenic on durum wheat in controlled assays (with 3 weeks delay after inoculation)? Please add this important information to material and method. Is this strain pathogenic on bread wheat in controlled inocultation. Please add this important information to material and method. Indeed, Zt strains could be either pathogenic to both wheat types or only one type.

We have added the information requested in the methods section. Specifically, the information is:

The Z.tritici isolate was obtained from a durum wheat cultivar.

We inoculated wheat cultivars (durum wheat: Svevo and Duilio cvs; bread wheat: Salamandra cv) with Zt 18258 in greenhouse. After 4 days, we cut from seedlings some leaf segments and put them on agar in Petri dishes at 20°C. Three weeks later, we observed a high presence of pycnidia.

Zt strain was pathogenic to both wheat species.

- Kernels of two Italian commercial varieties of durum wheat (Svevo and Duilio; Syngenta Italia and SIS società italiana sementi, respectively, ) of durum wheat (T. turgidum subsp. durum (Desf. Husn.), moderately susceptible to the SLBC, Reviewer comment: What is the level of susceptibility of these two Durum wheat cultivars to P nodorum on one side and Z tritici on the other side? It should be quite different, since resistance/susceptibility to these diseases are controlled by very different genetic factors. For example, which susceptibility genes to P nodorum are found in these cultivars, if known?

P.nodorum

We previously inoculated Svevo and Duilio cultivars with various P.nodorum isolates at the seedlings stage in greenhouse. We observed a range of responses from moderately resistant to susceptible for both the species (Iori et al., 2011). We also observed that Svevo and Duilio were moderately susceptible/susceptible if artificially inoculated in field (Iori et al., 2015a; 2015b). In the present study, in phytotron Svevo showed a susceptible behavior whereas Duilio a moderately resistant one and in field Svevo showed a moderately susceptible behavior.

  1. tritici

This is our first study on Svevo durum wheat artificially inoculated in field with Z tritici. Svevo was moderately susceptible to the pathogen.

To my knowledge, we still lack information on the susceptibility genes in these cultivars. This will be a key aspect to address in the following years.

  • Iori A., L’Aurora A., Niglio 2011. Behavior of wheat cultivars in organic farming tested at the seedlingstage with Stagonospora nodorum. Annual wheat newsletter 57: 37-39. 
  • Iori A., Scala V., Cesare D., Pinzari, Reverberi M., D’Egidio M.G., Corrado F., Fabbri A.A., Serranti S. 2015a . Hyperspectral and molecular analysis of stagonospora nodorum blotch in durum wheat. Doi: 10.1007/s10658-014-0571-x. European Journal of Plant Pathology 141: 689-702.
  • Iori A., Scala V., Fornara M., Reverberi M., Pietricola, Farina V., Mazzon V., Cristofori C., Quaranta F. 2015b. Stagonospora nodorum blotch on durum and common wheat varieties in organic farming: preliminary phytopathological and quali-quantitative data. ICC/AISTEC Conference “Grains for Feeding the World”. Milano 1 – 3 July 2015. Conference Proceedings ISBN: 978-88-906680-4-3 pag.137 – 140.

- Each plot in the two fields contained n = 250 plants (plants were fewer than seed probably for the considerable presence of birds at the time of sowing). The trials were performed in three plots, each containing all treatments per thesis (mock, T, Inf, T+Inf). Inoculation for the Artificial contamination in the field plots was prepared as indicated in “Preparation of fungal inoculum.” Reviewer comment: It seems that the same P nodorum and Z tritici strains have been used for Phytotron and Field trials, but this is not clearly stated. Please add the name of the strain used in all the Figure , Tables, so there is no doubt about it.

We inserted in table and figs and elsewhere the number of the isolates that, as you suggested, have been used for both controlled and field trials

Result Section

- Table 2. Disease severity at the second leaf stage according to Liu’s scale (0-5) in the durum wheat varieties (Svevo and Duilio) inoculated with P. nodorum or Z. tritici. Mock: without Tramesan, non-inoculated. T: treated with Tramesan; Tr+Inf: treated with Tramesan, inoculated with the pathogenInf: inoculated with the pathogen. Fisher test on n=48 × 2 biological repetitions. Reviewer comment: Add the time at which symptoms were scored, ie 7 days, and give the reference of the scale used in legend.

The time at which symptoms were scored, ie 7 days, was now added in the Table legend

- Comments on Table 2

Results showed that plants treated with Tramesan had fewer symptoms compared to the untreated ones, and that Tramesan reduces disease severity by P. nodorum (up to 75%) more than that by Z. tritici (up to 65%), even if these differences are not statistically significant (p=0.06). Reviewer comment: changesee abstract

The comment in the results has been added as suggested

- Comments on Figure 1

By quantifying the fungal DNA within leaf tissues at 0, 3, and 9 dai (Fig. 1 a,b), we found that the pathogen load dramatically decreased under Tramesan treatment by up to 99% for P. nodorum, and up to 75% for Z. triticiReviewer comment: Please give the statistical significance of the comparison in leaf fungal DNA content between Inf vs Inf+T for each cultivar x strain combination. This is not clear actually in Figure 1, as it looks like Authors have compared the two diseases for infection on one side and infection + T on the other side. Authors could use a specific table if it is easier than statistics labels on Figure 1, as these data are very important for demonstrating that Tramesan strongly reduces infection, as assayed by quantifying fungal growth.

The novel figure 1 has been modified according to the next suggestion, i.e. now it shows the infection for each strain/cultivar. Regarding the letters related to some bars, they identified specific groups after the ANOVA and Fisher Test. In the graphs only “b” or “c” appear, since every other bar is grouped under “a”. Thus, we chose to not show the “a” for the sake of legibility.

- Figure 1 Reviewer comment: It would be better to group together the kinetic of infection for each strain/cultivar pair, since it is quite different quantitatively for each interaction (really not the same scale). This would make Figure 1 more easy to follow.

As indicated above, we inserted a novel, modified, Figure 1

- Furthermore, we analyzed the yield of the harvested grain, proving that the Tramesan treatment (5.29 ± 0.05 t/ha) did not affect it compared to the untreated plots (5.28 ± 0.08 t/ha). Reviewer comment: better to say: In addition, Tramesan treatment did not affect grain yields (5.29 ± 0.05 t/ha) compared to untreated plots (5.28 ± 0.08 t/ha) that only displayed limited SLBC leaf infection in our assay.

Adjusted as suggested

- The measurement of the content of defense-related hormones such as SA and JA….Reviewer comment: Better to say The assay of SA and JA levels in infected leaves …

Adjusted as suggested

- Figure 3: Reviewer comment: it would be nice to add value for SA and JA relative abundance just under the each box.

The figure 3 has been modified according to your comment

- Figure 4 has been really improved. Reviewer comment: It could be nice to use colored lines (Red, Blue) to highlight infection + T compared to infection (black).

The figure 4 has been modified according to your comment

Discussion

- we posit Reviewer comment: better to use we propose

Posit changed into propose
